# Dimension-Free Bounds for the Union-Closed Sets Conjecture

**DOI:** 10.3390/e25050767

**Published:** 2023-05-08

**Authors:** Lei Yu

**Affiliations:** School of Statistics and Data Science, The Key Laboratory of Pure Mathematics and Combinatorics (LPMC), Key Laboratory for Medical Data Analysis and Statistical Research of Tianjin (KLMDASR), and Laboratory for Economic Behaviors and Policy Simulation (LEBPS), Nankai University, Tianjin 300071, China; leiyu@nankai.edu.cn

**Keywords:** union-closed sets conjecture, information-theoretic method, coupling

## Abstract

The union-closed sets conjecture states that, in any nonempty union-closed family F of subsets of a finite set, there exists an element contained in at least a proportion 1/2 of the sets of F. Using an information-theoretic method, Gilmer recently showed that there exists an element contained in at least a proportion 0.01 of the sets of such F. He conjectured that their technique can be pushed to the constant $\frac{3-\sqrt{5}}{2}$ which was subsequently confirmed by several researchers including Sawin. Furthermore, Sawin also showed that Gilmer’s technique can be improved to obtain a bound better than $\frac{3-\sqrt{5}}{2}$ but this new bound was not explicitly given by Sawin. This paper further improves Gilmer’s technique to derive new bounds in the optimization form for the union-closed sets conjecture. These bounds include Sawin’s improvement as a special case. By providing cardinality bounds on auxiliary random variables, we make Sawin’s improvement computable and then evaluate it numerically, which yields a bound approximately 0.38234, slightly better than $\frac{3-\sqrt{5}}{2}\approx 0.38197$.

## 1. Introduction

This paper concerns the union-closed sets conjecture which is described in the information-theoretic language as follows. For that purpose, every set B⊆[n]:={1,2,…,n} is uniquely described by an *n*-length sequence $x^{n}:=\left(x_{1},x_{2},\ldots ,x_{n}\right)\in \Omega^{n}$ with Ω:={0,1} such that $x_{i}=1$ if i∈B and $x_{i}=0$ otherwise. So, a family F of subsets of [n] uniquely corresponds to a subset $A\subseteq \Omega^{n}$. Denote the (element-wise) OR operation for two finite Ω-valued sequences as $x^{n}\vee y^{n}:={\left(x_{i}\vee y_{i}\right)}_{i\in [n]}$ with $x^{n},y^{n}\in \Omega^{n}$, where ∨ is the OR operation. The family F is closed under the union operation (i.e., F∪G∈F,∀F,G∈F) if and only if the corresponding set $A\subseteq \Omega^{n}$ is closed under the OR operation (i.e., $x^{n}\vee y^{n}\in A,\forall x^{n},y^{n}\in A$).

Let $A\subseteq \Omega^{n}$ be closed under the OR operation. Let $X^{n}:=\left(X_{1},X_{2},\ldots ,X_{n}\right)$ be a random vector uniformly distributed on *A* and denote $P_{X^{n}}=\mathrm{Unif}\left(A\right)$ as its distribution (or probability mass function, PMF). We are interested in estimating
$$p_{A}:=\max_{i\in [n]}P_{X_{i}}\left(1\right)$$
where $P_{X_{i}}$ is the distribution of $X_{i}$ and, hence, $P_{X_{i}}\left(1\right)$ is the proportion of the sets containing the element *i* among all sets in F. Frankl made the following conjecture.

**Conjecture 1** (Frankl Union-Closed Sets Conjecture). *$p_{A}\geq 1/2$ for any OR-closed set A.*

This conjecture equivalently states that, for any union-closed family F, there exists an element contained in at least a proportion 1/2 of the sets of F. Since the union-closed conjecture was posed by Peter Frankl in 1979, it has attracted a great deal of research interest; see, e.g., [1,2,3,4,5]. We refer readers to the survey paper [6] for more details. Gilmer [7] made a breakthrough recently, showing that this conjecture holds with constant 0.01. His method used a clever idea from information theory in which two independent random vectors were constructed. It was conjectured by Gilmer that his method can improve the constant to $\frac{3-\sqrt{5}}{2}$, which is now confirmed by several groups of researchers [8,9,10,11]. This constant is shown to be the best for an approximate version of the union-closed sets problem [9]. Moreover, Sawin [8] further developed Gilmer’s idea by allowing the two random vectors to depend on each other. In fact, the same idea was previously used by the present author in several works [12,13,14]. By this technique, Sawin [8] showed that the constant can be improved to a value that is strictly larger than $\frac{3-\sqrt{5}}{2}$. However, without cardinality bounds on auxiliary random variables, Sawin’s constant is difficult to compute, hence the accurate value of this improved constant is not explicitly given in [8].

The present paper further develops Gilmer’s (or Sawin’s) technique to derive new constants (or bounds) in the optimization form for the union-closed sets conjecture. These bounds include Sawin’s improvement as a special case. By providing cardinality bounds on auxiliary random variables, we make Sawin’s improvement computable and then evaluate it numerically which yields a bound approximately 0.38234, slightly better than $\frac{3-\sqrt{5}}{2}\approx 0.38197$.

## 2. Main Results

To state our result, we need to introduce some notations. Since we only consider distributions on finite alphabets, we do not distinguish between the terms “distributions” and “probability mass functions”. For a pair of distributions $\left(P_{X},P_{Y}\right)$, a coupling of $\left(P_{X},P_{Y}\right)$ is a joint distribution $P_{XY}$ whose marginals are, respectively, $P_{X},P_{Y}$. For a distribution $P_{X}$ defined on a finite alphabet X, a coupling $P_{XX^{\prime }}$ of $\left(P_{X},P_{X}\right)$ is called symmetric if $P_{XX^{\prime }}\left(x,y\right)=P_{XX^{\prime }}\left(y,x\right)$ for all x,y∈X. Denote $C_{s}\left(P_{X}\right)$ as the set of symmetric couplings of $\left(P_{X},P_{X}\right)$. Denote $\delta_{x}$ as the Dirac measure with a single atom at *x*. That is, the PMF of this measure takes the value 1 at *x* and takes the value 0 at other points.

For a joint distribution $P_{XY}$, the (Pearson) correlation coefficient between $\left(X,Y\right)∼P_{XY}$ is defined by
$$\rho_{p}\left(X;Y\right):=\left\{\begin{array}{cc} \frac{\mathrm{Cov}(X,Y)}{\sqrt{\mathrm{Var}(X)\mathrm{Var}(Y)}}, & \mathrm{Var}(X)\mathrm{Var}(Y)>0\\ 0, & \mathrm{Var}(X)\mathrm{Var}(Y)=0 \end{array}\right..$$
The maximal correlation between $\left(X,Y\right)∼P_{XY}$ is defined by
$$\begin{array}{cc} \rho_{m}\left(X;Y\right) & :=\underset{f,g}{\sup}\rho_{p}\left(f\left(X\right);g\left(Y\right)\right)\\ \; & =\underset{f,g}{\sup}\left\{\begin{array}{cc} \frac{\mathrm{Cov}(f(X),g(Y))}{\sqrt{\mathrm{Var}(f(X))\mathrm{Var}(g(Y))}}, & \mathrm{Var}(f(X))\mathrm{Var}(g(Y))>0\\ 0, & \mathrm{Var}(f(X))\mathrm{Var}(g(Y))=0 \end{array}\right., \end{array}$$
where the supremum is taken over all pairs of real-valued functions $\left(f,g\right)$ such that Var(f(X)),Var(g(Y))<∞. Note that $\rho_{m}\left(X;Y\right)\in \left[0,1\right]$ and, moreover, $\rho_{m}\left(X;Y\right)=0$ if and only if X,Y are independent. Moreover, $\rho_{m}\left(X;Y\right)$ is equal to the second largest singular value of the matrix ${\left[\frac{P_{XY}\left(x,y\right)}{\sqrt{P_{X}\left(x\right)P_{Y}\left(y\right)}}\right]}_{\left(x,y\right)}$; see, e.g., [15]. Clearly, the largest singular value of the matrix ${\left[\frac{P_{XY}\left(x,y\right)}{\sqrt{P_{X}\left(x\right)P_{Y}\left(y\right)}}\right]}_{\left(x,y\right)}$ is equal to 1 with corresponding eigenvectors ${\left(\sqrt{P_{X}\left(x\right)}\right)}_{x}$ and ${\left(\sqrt{P_{Y}\left(y\right)}\right)}_{y}$.

Denote for p,q,ρ∈[0,1],
$$\begin{array}{cc} z_{1} & :=pq-\rho \sqrt{p(1-p)q(1-q)}\\ z_{2} & :=pq+\rho \sqrt{p(1-p)q(1-q)} \end{array}$$
and
(1)$$\varphi \left(\rho ,p,q\right):=\mathrm{median}\left\{\max\left\{p,q,p+q-z_{2}\right\},1/2,\min\left\{p+q,p+q-z_{1}\right\}\right\},$$
where median(A) denotes the median value of elements in a multiset *A*. We regard the set in (Equation 1) as a multiset which means median{a,a,b}=a. Denote $h\left(a\right)=-a\log_{2}a-\left(1-a\right)\log_{2}\left(1-a\right)$ for a∈[0,1] as the binary entropy function. Define for t>0,
(2)$$\Gamma \left(t\right):=\underset{P_{\rho }}{\sup}\underset{P_{p}:Eh\left(p\right)>0,Ep\leq t}{\inf}E_{\rho }\left[\underset{P_{pq}\in C_{s}\left(P_{p}\right):\rho_{m}\left(p;q\right)\leq \rho }{\inf}\frac{E_{p,q}h\left(\varphi \left(\rho ,p,q\right)\right)}{Eh(p)}\right],$$
where the supremum over $P_{\rho }$ and the infimum over $P_{p}$ are both taken over all finitely supported probability distributions on [0,1].

Our main results are as follows.

**Theorem 1.** 
*If Γ(t)>1 for some t∈(0,1/2), then $p_{A}\geq t$ for any OR-closed $A\subseteq \Omega^{n}$ (i.e., for any union-closed family F, there exists an element contained in at least a proportion t of the sets of F).*


The proof of Theorem 1 is given in Section 2 by using a technique based on coupling and entropy. It is essentially the same as the technique used by Sawin [8]. Prior to Sawin’s work, such a technique was used by the present author in several works; see [12,13,14].

Equivalently, Theorem 1 states that $p_{A}\geq t_{\sup}$ for any OR-closed $A\subseteq \Omega^{n}$, where $t_{\sup}:=\sup\left\{t\in \left(0,1/2\right):\Gamma \left(t\right)>1\right\}.$ To compute Γ(t) numerically, it is required to upper bound the cardinality of the support of $P_{p}$ in the outer infimum in (Equation 2) since, otherwise, infinitely many parameters are needed to optimize. This is left to be done in a future work. We next provides a computable bound, which is a lower bound of Γ(t), instead Γ(t) itself.

If we choose $P_{\rho }=\delta_{0}$, then Theorem 1 implies Gilmer’s bound in [7] since, for this case, the couplings constructed in the proof of Theorem 1 (given in the next section) turn out to be independent, coinciding with Gilmer’s construction. On the other hand, if we choose $P_{\rho }=\delta_{1}$, then the couplings constructed in our proof are arbitrary. In fact, we can make a choice of $P_{\rho }$ better than these two special cases. As suggested by Sawin [8], we can choose $P_{\rho }=\left(1-\alpha \right)\delta_{0}+\alpha \delta_{1}$ which in fact leads to an optimization over mixtures of independent couplings and arbitrary couplings. This final choice yields the following bound.

Substituting ρ=0 and 1, respectively, into φ(ρ,p,q) yields
(3)$$\begin{array}{cc} \varphi (0,p,q) & =p+q-pq,\;\;\;\;\;\;\;\; \end{array}$$
(4)$$\begin{array}{cc} \varphi (1,p,q) & =\mathrm{median}\left\{\max\{p,q\},1/2,p+q\right\}, \end{array}$$
where, in the evaluation of φ(1,p,q), the following facts were used: (1)
$$p+q-pq-\sqrt{p(1-p)q(1-q)}\leq \max\left\{p,q\right\}$$
for all p,q∈[0,1]; (2) if p+q≤1, then
$$p+q-pq+\sqrt{p(1-p)q(1-q)}\geq p+q,$$
and otherwise,
$$1/2<\max\left\{p,q\right\}\leq p+q-pq+\sqrt{p(1-p)q(1-q)}.$$
By defining
$$\begin{array}{cc} g(P_{pq},\alpha ) & :=\left(1-\alpha \right)E_{\left(p,q\right)∼P_{p}^{⊗2}}\left[h\left(p+q-pq\right)\right]+\alpha E_{\left(p,q\right)∼P_{pq}}\left[h\left(\varphi \left(1,p,q\right)\right)\right] \end{array}$$
and substituting $P_{\rho }=\left(1-\alpha \right)\delta_{0}+\alpha \delta_{1}$ into Theorem 1, one obtains the following simpler bound.

**Proposition 1.** 
*For t∈(0,1/2),*

(5)
$$\Gamma \left(t\right)\geq \hat{\Gamma }\left(t\right):=\underset{\alpha \in [0,1]}{\sup}\underset{\mathrm{symmetric}P_{pq}:Eh\left(p\right)>0}{\inf}\frac{g(P_{pq},\alpha )}{Eh(p)},$$

*where the infimum is taken over all distributions $P_{pq}$ of the form $\left(1-\beta \right)Q_{a_{1},a_{2}}+\beta Q_{b_{1},b_{2}}$ with*

(6)
$$0\leq a:=\frac{a_{1}+a_{2}}{2}\leq t<b:=\frac{b_{1}+b_{2}}{2}\leq 1$$

*and β=0 or $\beta =\frac{t-a}{b-a}>0$ such that Eh(p)>0. (Note that Eh(p)=0 if and only if $P_{pq}$ is a convex combination of $\delta_{\left(0,0\right)}$, $\delta_{\left(0,1\right)}$, $\delta_{\left(1,0\right)}$, and $\delta_{\left(1,1\right)}$.) Here,*

(7)
$$Q_{x,y}:=\frac{1}{2}\delta_{\left(x,y\right)}+\frac{1}{2}\delta_{\left(y,x\right)}$$

*with $\delta_{\left(x,y\right)}$ denoting the Dirac measure at (x,y) (whose PMF takes the value *1* at (x,y) and takes the value *0* at other points).*


As a consequence of the two results above, we have the following corollary.

**Corollary 1.** 
*If $\hat{\Gamma }\left(t\right)>1$ for some t∈(0,1/2), then $p_{A}\geq t$ for any OR-closed $A\subseteq \Omega^{n}$.*


The proof of Corollary 1 is given in Section 3.

The lower bound in (Equation 5) without the cardinality bound on the support of $P_{pq}$ was given by Sawin [8], which was used to show $p_{A}>\frac{3-\sqrt{5}}{2}.$ However, thanks to the cardinality bound, we can numerically compute the best bound on $p_{A}$ that can be derived using $\hat{\Gamma }\left(t\right)$. That is, $p_{A}\geq {\hat{t}}_{\sup}$ for any OR-closed $A\subseteq \Omega^{n}$, where ${\hat{t}}_{\sup}:=\sup\left\{t\in \left(0,1/2\right):\hat{\Gamma }\left(t\right)>1\right\}.$ Numerical results show that if we set α=0.035,t=0.38234, then the optimal $P_{pq}=\left(1-\beta \right)Q_{a,a}+\beta Q_{a,1}$ with a≈0.3300622 and β≈0.1560676 which leads to the lower bound $\hat{\Gamma }\left(t\right)\geq 1.00000889$. Hence, $p_{A}\geq 0.38234$ for any OR-closed $A\subseteq \Omega^{n}$. This is slightly better than the previous bound $\frac{3-\sqrt{5}}{2}\approx 0.38197$. The choice of (α,t) in our evaluation is nearly optimal. Our code can be found on the author’s homepage https://leiyudotscholar.wordpress.com/ (accessed on 1 May 2023.) More decimal places of Sawin’s bound (or equivalently, ${\hat{t}}_{\sup}$) were computed by Cambie in a concurrent work [16], i.e., $0.382345533366702\leq {\hat{t}}_{\sup}\leq 0.382345533366703$ which is attained by the choice α≈0.03560698136437784. This more precise evaluation can be also verified using our code above.

## 3. Proof of Theorem 1

Denote $H\left(X\right)=-\sum_{x}P_{X}\left(x\right)\log P_{X}\left(x\right)$ as the Shannon entropy of a random variable $X∼P_{X}$. Let $A\subseteq \Omega^{n}$ be closed under the OR operation. We assume |A|≥2. This is because Theorem 1 holds obviously for singletons *A*, since for this case, $p_{A}=1$. Let $P_{X^{n}}=\mathrm{Unif}\left(A\right)$. So, $H(X^{n})>0$ and, by the chain rule, $H\left(X^{n}\right)=\sum_{i=1}^{n}H\left(X_{i}|X^{i-1}\right)$.

If $P_{X^{n}Y^{n}}\in C_{s}\left(P_{X^{n}}\right)$, then $Z^{n}:=X^{n}\vee Y^{n}\in A$ a.s. where $\left(X^{n},Y^{n}\right)∼P_{X^{n}Y^{n}}$. So, we have
$$H\left(Z^{n}\right)\leq \log\left|A\right|=H\left(X^{n}\right).$$
We hence have
$$\underset{P_{X^{n}Y^{n}}\in C_{s}\left(P_{X^{n}}\right)}{\sup}\frac{H(Z^{n})}{H(X^{n})}\leq 1.$$

If $p_{A}\leq t$, then $P_{X_{i}}\left(1\right)\leq t,\forall i\in \left[n\right]$. Relaxing $P_{X^{n}}=\mathrm{Unif}\left(A\right)$ to arbitrary distributions such that $P_{X_{i}}\left(1\right)\leq t$, we obtain $\Gamma_{n}\left(t\right)\leq 1$ where
(8)$$\Gamma_{n}\left(t\right):=\underset{P_{X^{n}}:P_{X_{i}}\left(1\right)\leq t,\forall i}{\inf}\underset{P_{X^{n}Y^{n}}\in C_{s}\left(P_{X^{n}}\right)}{\sup}\frac{H(Z^{n})}{H(X^{n})}.$$
In other words, if given *t*, $\Gamma_{n}\left(t\right)>1$, then, by contradiction, $p_{A}>t$.

We next show that $\Gamma_{n}\left(t\right)\geq \Gamma \left(t\right)$ which implies Theorem 1. To this end, we need the following lemmas.

For two conditional distributions $P_{X|U},P_{Y|V}$, denote $C(P_{X|U},P_{Y|V})$ as the set of conditional distributions $Q_{XY|UV}$ such that their marginals satisfy $Q_{X|UV}=P_{X|U},Q_{Y|UV}=P_{Y|V}$. The conditional (Pearson) correlation coefficient of *X* and *Y* given *U* is defined by
$$\rho_{p}\left(X;Y|U\right)=\left\{\begin{array}{cc} \frac{E[\mathrm{cov}(X,Y|U)]}{\sqrt{E[\mathrm{var}(X|U)]}\sqrt{E[\mathrm{var}(Y|U)]}}, & E[\mathrm{var}(X|U)]E[\mathrm{var}(Y|U)]>0,\\ 0, & E[\mathrm{var}(X|U)]E[\mathrm{var}(Y|U)]=0. \end{array}\right.$$
The conditional maximal correlation coefficient of *X* and *Y* given *U* is defined by
$$\rho_{m}\left(X;Y|U\right)=\underset{f,g}{\sup}\rho_{p}\left(f\left(X,U\right);g\left(Y,U\right)|U\right),$$
where the supremum is taken over all real-valued functions f(x,u),g(y,u) such that E[var(f(X,U)|U)], E[var(g(Y,U)|U)]<∞. It has been shown in [17] that
$$\rho_{m}\left(X;Y|U\right)=\underset{u:P_{U}\left(u\right)>0}{\sup}\rho_{m}\left(X;Y|U=u\right),$$
where $\rho_{m}\left(X;Y|U=u\right)=\rho_{m}\left(X^{\prime };Y^{\prime }\right)$ with $\left(X^{\prime },Y^{\prime }\right)∼P_{XY|U=u}$.

**Lemma 1** (Product Construction of Couplings). *Lemma 9 in [12], Corollary 3 in [17], and Lemma 6 in [18] For any conditional distributions $P_{X_{i}|X^{i-1}},\;P_{Y_{i}|Y^{i-1}},\;i\in \left[n\right]$ and any*
$$Q_{X_{i}Y_{i}|X^{i-1}Y^{i-1}}\in C\left(P_{X_{i}|X^{i-1}},P_{Y_{i}|Y^{i-1}}\right),\forall i\in \left[n\right],$$
*it holds that*
(9)$$\begin{array}{cc} \prod_{i=1}^{n}Q_{X_{i}Y_{i}|X^{i-1}Y^{i-1}} & \in C\left(\prod_{i=1}^{n}P_{X_{i}|X^{i-1}},\prod_{i=1}^{n}P_{Y_{i}|Y^{i-1}}\right). \end{array}$$
*Moreover, for $\left(X^{n},Y^{n}\right)∼\prod_{i=1}^{n}Q_{X_{i}Y_{i}|X^{i-1}Y^{i-1}}$, it holds that*
(10)$$\rho_{m}\left(X^{n};Y^{n}\right)=\max_{i\in [n]}\rho_{m}\left(X_{i};Y_{i}|X^{i-1},Y^{i-1}\right).$$

For a conditional distribution $P_{X|U}$ defined on finite alphabets, a conditional coupling $P_{XX^{\prime }|UU^{\prime }}$ of $\left(P_{X|U},P_{X|U}\right)$ is called symmetric if $P_{XX^{\prime }|UU^{\prime }}\left(x,y|u,v\right)=P_{XX^{\prime }|UU^{\prime }}\left(y,x|v,u\right)$ for all x,y∈X,u,v∈U. Denote $C_{s}\left(P_{X|U}\right)$ as the set of symmetric conditional couplings of $\left(P_{X|U},P_{X|U}\right)$. Applying the lemma above to symmetric couplings, we have that if couplings $Q_{X_{i}Y_{i}|X^{i-1}Y^{i-1}}\in C_{s}\left(P_{X_{i}|X^{i-1}}\right)$ satisfy $\rho_{m}\left(X_{i};Y_{i}|X^{i-1},Y^{i-1}\right)\leq \rho$ for some ρ>0, then
$$\begin{array}{cc} \prod_{i=1}^{n}Q_{X_{i}Y_{i}|X^{i-1}Y^{i-1}} & \in C_{s}\left(\prod_{i=1}^{n}P_{X_{i}|X^{i-1}}\right),\\ \rho_{m}\left(X^{n};Y^{n}\right) & \leq \rho , \end{array}$$
with $\left(X^{n},Y^{n}\right)∼\prod_{i=1}^{n}Q_{X_{i}Y_{i}|X^{i-1}Y^{i-1}}$. We hence have that, for any ρ∈[0,1],
(11)$$\begin{array}{cc} \; & \underset{\begin{array}{c} P_{X^{n}Y^{n}}\in C_{s}\left(P_{X^{n}}\right):\\ \rho_{m}\left(X^{n};Y^{n}\right)\leq \rho \end{array}}{\sup}H\left(Z^{n}\right)\\ \; & \geq \underset{\begin{array}{c} P_{X^{n-1}Y^{n-1}}\in C_{s}\left(P_{X^{n-1}}\right):\\ \rho_{m}\left(X^{n-1};Y^{n-1}\right)\leq \rho \end{array}}{\sup}\left(H\left(Z^{n-1}\right)+\underset{\begin{array}{c} P_{X_{n}Y_{n}|X^{n-1}Y^{n-1}}\in C_{s}\left(P_{X_{n}|X^{n-1}}\right):\\ \rho_{m}\left(X_{n};Y_{n}|X^{n-1},Y^{n-1}\right)\leq \rho \end{array}}{\sup}H\left(Z_{n}|Z^{n-1}\right)\right)\\ \; & \geq \underset{\begin{array}{c} P_{X^{n-1}Y^{n-1}}\in C_{s}\left(P_{X^{n-1}}\right):\\ \rho_{m}\left(X^{n-1};Y^{n-1}\right)\leq \rho \end{array}}{\sup}H\left(Z^{n-1}\right)\\ \; & \;+\underset{\begin{array}{c} P_{X^{n-1}Y^{n-1}}\in C_{s}\left(P_{X^{n-1}}\right):\\ \rho_{m}\left(X^{n-1};Y^{n-1}\right)\leq \rho \end{array}}{\inf}\underset{\begin{array}{c} P_{X_{n}Y_{n}|X^{n-1}Y^{n-1}}\in C_{s}\left(P_{X_{n}|X^{n-1}}\right):\\ \rho_{m}\left(X_{n};Y_{n}|X^{n-1},Y^{n-1}\right)\leq \rho \end{array}}{\sup}H\left(Z_{n}|Z^{n-1}\right)\\ \; & \geq \ldots \ldots \\ \; & \geq \sum_{i=1}^{n}\underset{\begin{array}{c} P_{X^{i-1}Y^{i-1}}\in C_{s}\left(P_{X^{i-1}}\right):\\ \rho_{m}\left(X^{i-1};Y^{i-1}\right)\leq \rho \end{array}}{\inf}\underset{\begin{array}{c} P_{X_{i}Y_{i}|X^{i-1}Y^{i-1}}\in C_{s}\left(P_{X_{i}|X^{i-1}}\right):\\ \rho_{m}\left(X_{i};Y_{i}|X^{i-1},Y^{i-1}\right)\leq \rho \end{array}}{\sup}H\left(Z_{i}|Z^{i-1}\right), \end{array}$$
where the first inequality above follows by Lemma 1 and the chain rule for entropies. In fact, in the derivation above, the *i*-th distribution $P_{X_{i}Y_{i}|X^{i-1}Y^{i-1}}$ is chosen as a greedy coupling in the sense that it only maximizes the *i*-th objective function $H(Z_{i}|Z^{i-1})$, regardless of other $H(Z_{j}|Z^{j-1})$ with j>i (although it indeed affects their values).

By the fact that conditioning reduces entropy, it holds that
$$H\left(Z_{i}|Z^{i-1}\right)\geq H\left(Z_{i}|X^{i-1},Y^{i-1}\right).$$
Denote
(12)$$\begin{array}{cc} g_{i}\left(P_{X^{i-1}},\rho \right):= & \underset{\begin{array}{c} P_{X^{i-1}Y^{i-1}}\in C_{s}\left(P_{X^{i-1}}\right):\\ \rho_{m}\left(X^{i-1};Y^{i-1}\right)\leq \rho \end{array}}{\inf}\underset{\begin{array}{c} P_{X_{i}Y_{i}|X^{i-1}Y^{i-1}}\in C_{s}\left(P_{X_{i}|X^{i-1}}\right):\\ \rho_{m}\left(X_{i};Y_{i}|X^{i-1},Y^{i-1}\right)\leq \rho \end{array}}{\sup}H\left(Z_{i}|X^{i-1},Y^{i-1}\right). \end{array}$$
Then, the expression at the right-hand side of (Equation 11) is further lower bounded by $\sum_{i=1}^{n}g_{i}\left(P_{X^{i-1}},\rho \right)$. Combing this with (Equation 8) and (Equation 11), and by noting that ρ∈[0,1] is arbitrary, we obtain that
(13)$$\begin{array}{cc} \Gamma_{n}\left(t\right) & \geq \underset{P_{X^{n}}:P_{X_{i}}\left(1\right)\leq t,\forall i}{\inf}\frac{\sup_{\rho \in [0,1]}\sum_{i=1}^{n}g_{i}\left(P_{X^{i-1}},\rho \right)}{\sum_{i=1}^{n}H\left(X_{i}|X^{i-1}\right)}\\ \; & =\underset{P_{X^{n}}:P_{X_{i}}\left(1\right)\leq t,\forall i}{\inf}\frac{\sup_{P_{\rho }}E_{P_{\rho }}\sum_{i=1}^{n}g_{i}\left(P_{X^{i-1}},\rho \right)}{\sum_{i=1}^{n}H\left(X_{i}|X^{i-1}\right)}\\ \; & \geq \underset{P_{\rho }}{\sup}\underset{P_{X^{n}}:P_{X_{i}}\left(1\right)\leq t,\forall i}{\inf}\frac{\sum_{i=1}^{n}E_{P_{\rho }}g_{i}\left(P_{X^{i-1}},\rho \right)}{\sum_{i=1}^{n}H\left(X_{i}|X^{i-1}\right)}\\ \; & \geq \underset{P_{\rho }}{\sup}\underset{P_{X^{n}}:P_{X_{i}}\left(1\right)\leq t,\forall i}{\inf}\min_{i\in \left[n\right]:H\left(X_{i}|X^{i-1}\right)>0}\frac{E_{P_{\rho }}g_{i}\left(P_{X^{i-1}},\rho \right)}{H(X_{i}|X^{i-1})}\\ \; & \geq \underset{P_{\rho }}{\sup}\underset{P_{X^{j}}:H\left(X_{j}|X^{j-1}\right)>0,P_{X_{j}}\left(1\right)\leq t}{\inf}\frac{E_{P_{\rho }}g_{j}\left(P_{X^{j-1}},\rho \right)}{H(X_{j}|X^{j-1})}, \end{array}$$
where
(Equation 13) follows since $\frac{a+b}{c+d}\geq \min\left\{\frac{a}{c},\frac{b}{d}\right\}$ for a,b≥0,c,d>0, and $H(X_{i}|X^{i-1})=0$ implies $X_{i}$ is a deterministic function of $X^{i-1}$ and, hence, $g_{i}\left(P_{X^{i-1}},\rho \right)=0$;The index *j* in the last line is the optimal *i* attaining the minimum in (Equation 13).
Denote $X=X_{j},Y=Y_{j},U=X^{j-1},V=Y^{j-1}$, and Z=X∨Y. Then,
(14)$$\begin{array}{cc} \Gamma_{n}\left(t\right) & \geq \underset{P_{\rho }}{\sup}\underset{P_{UX}:H\left(X|U\right)>0,P_{X}\left(1\right)\leq t}{\inf}E_{P_{\rho }}\left[\underset{\begin{array}{c} P_{UV}\in C_{s}\left(P_{U}\right):\\ \rho_{m}\left(U;V\right)\leq \rho \end{array}}{\inf}\underset{\begin{array}{c} P_{XY|UV}\in C_{s}\left(P_{X|U}\right):\\ \rho_{m}\left(X;Y|U,V\right)\leq \rho \end{array}}{\sup}\frac{H(Z|U,V)}{H(X|U)}\right]. \end{array}$$

We next further simplify the lower bound in (Equation 14). Denote
(15)$$p=P_{X|U}\left(1|U\right),q=P_{Y|V}\left(1|V\right),r=P_{XY|UV}\left(1,1|U,V\right).$$
So,
$$P_{XY|UV}\left(\cdot |U,V\right)=\left[\begin{array}{cc} 1+r-p-q & q-r\\ p-r & r \end{array}\right]$$
with
max{0,p+q−1}≤r≤min{p,q}.
Note that
(16)$$\begin{array}{cc} \rho_{m}\left(X;Y|U,V\right) & =\underset{u,v:P_{UV}\left(u,v\right)>0}{\sup}\rho_{m}\left(X_{u};Y_{v}\right)\\ \; & =\underset{u,v:P_{UV}\left(u,v\right)>0}{\sup}\left|\rho_{p}\left(X_{u};Y_{v}\right)\right|\\ \; & =\underset{u,v:P_{UV}\left(u,v\right)>0}{\sup}\frac{\left|r-pq\right|}{\sqrt{p(1-p)q(1-q)}}, \end{array}$$
where $\left(X_{u},Y_{v}\right)∼P_{XY|U=u,V=v}$, $\rho_{p}$ denotes the Pearson correlation coefficient and (Equation 16) follows since the maximal correlation coefficient between two binary random variables is equal to the absolute value of the Pearson correlation coefficient between them; see, e.g., [19]. So, $\rho_{m}\left(X;Y|U,V\right)\leq \rho$ is equivalent to $\frac{\left|r-pq\right|}{\sqrt{p(1-p)q(1-q)}}\leq \rho$ a.s. and also equivalent to $z_{1}\leq r\leq z_{2}$ a.s.

The inner supremum in (Equation 14) can be rewritten as
$$\begin{array}{cc} \; & \underset{P_{XY|UV}\in C_{s}\left(P_{X|U}\right):\rho_{m}\left(X;Y|U,V\right)\leq \rho }{\sup}H\left(Z|U,V\right)\\ \; & =E_{p,q}\underset{\max\left\{0,p+q-1,z_{1}\right\}\leq r\leq \min\left\{p,q,z_{2}\right\}}{\sup}h\left(p+q-r\right). \end{array}$$
By the fact that *h* is increasing on [0,1/2] and decreasing on [1/2,1], it holds that the optimal *r* attaining the supremum in the last line above, denoted by $r^{*}$, is the median of $\max\{0,p+q-1,z_{1}\}$, p+q−1/2, and $\min\{p,q,z_{2}\}$, which implies
$$p+q-r^{*}=\varphi \left(\rho ,p,q\right).$$
Recall the definition of φ in (Equation 1). So, the inner supremum in (Equation 14) is equal to $\frac{E_{p,q}h\left(\varphi \left(\rho ,p,q\right)\right)}{Eh(p)}$.

We make the following observations. Firstly,
$$\begin{array}{cc} H(X|U) & =Eh(p),\\ P_{X}\left(1\right) & =Ep. \end{array}$$
Secondly, by the definition of maximal correlation, $\rho_{m}\left(p;q\right)\leq \rho_{m}\left(U;V\right)$ holds (which is known as the data processing inequality) since p,q are, respectively, functions of U,V; see (Equation 15). Lastly, observe that $P_{UV}$ is symmetric and p,q are obtained from U,V via the same function $P_{X|U}\left(1|\cdot \right)$ (since $P_{X|U}=P_{Y|V}$ holds by the symmetry of $P_{XY|UV}$). Hence, $P_{pq}$ is symmetric as well. Substituting all of these into (Equation 14) yields $\Gamma_{n}\left(t\right)\geq \Gamma \left(t\right)$. □

## 4. Proof of Proposition 1

By choosing $P_{\rho }=\left(1-\alpha \right)\delta_{0}+\alpha \delta_{1}$ in (Equation 2), we obtain
$$\Gamma \left(t\right)\geq \underset{\alpha \in [0,1]}{\sup}\underset{\mathrm{symmetric}P_{pq}:Eh\left(p\right)>0,Ep\leq t}{\inf}\frac{g(P_{pq},\alpha )}{Eh(p)}.$$
Note that $P_{pq}\mapsto g\left(P_{pq},\alpha \right)$ is concave, since, by Lemma 5 in [10]$P_{p}\mapsto E_{\left(p,q\right)∼P_{p}^{⊗2}}h\left(p+q-pq\right)$ is concave, and $P_{pq}\mapsto P_{p}$ is linear.

Let *B* be a finite subset of [0,1]. Let $P_{B}$ be the set of symmetric distributions $P_{pq}$ concentrated on $B^{2}$ such that Ep≤t. By the Krein–Milman theorem, $P_{B}$ is equal to the closed convex hull of its extreme points. These extreme points are of the form $\left(1-\beta \right)Q_{a_{1},a_{2}}+\beta Q_{b_{1},b_{2}}$ with 0≤a≤t<b≤1 and β=0 or $\frac{t-a}{b-a}$; recall the definitions $a:=\frac{a_{1}+a_{2}}{2},b:=\frac{b_{1}+b_{2}}{2}$, and $Q_{x,y}:=\frac{1}{2}\delta_{\left(x,y\right)}+\frac{1}{2}\delta_{\left(y,x\right)}$ in (Equation 6) and (Equation 7). By Carathéodory’s theorem, it is easy to see that the convex hull of these extreme points is closed (in the weak topology or, equivalently, in the relative topology on the probability simplex). So, every $P_{pq}$ supported on a finite set $B^{2}\subseteq {\left[0,1\right]}^{2}$ such that Ep≤t is a convex combination of the extreme points above, i.e., $P_{pq}=\sum_{i=1}^{k}\gamma_{i}Q_{i}$ where $Q_{i},i\in \left[k\right]$ are extreme points, and $\gamma_{i}>0$ and $\sum_{i=1}^{k}\gamma_{i}=1$. For this distribution,
$$\begin{array}{cc} \frac{g(P_{pq},\alpha )}{Eh(p)} & =\frac{g(\sum_{i=1}^{k}\gamma_{i}Q_{i},\alpha )}{\sum_{i=1}^{k}\gamma_{i}E_{Q_{i}}h\left(p\right)}\\ \; & \geq \frac{\sum_{i=1}^{k}\gamma_{i}g\left(Q_{i},\alpha \right)}{\sum_{i=1}^{k}\gamma_{i}E_{Q_{i}}h\left(p\right)}\\ \; & \geq \min_{i:E_{Q_{i}}h\left(p\right)>0}\frac{g(Q_{i},\alpha )}{E_{Q_{i}}h\left(p\right)} \end{array}$$
where, in the last line, the constraint $E_{Q_{i}}h\left(p\right)>0$ is posed since $E_{Q_{i}}h\left(p\right)=0$ implies $Q_{i}=\delta_{\left(0,0\right)}$ (note that t<1/2) and, hence, $g(Q_{i},\alpha )=0$.

Therefore,
(17)$$\Gamma \left(t\right)\geq \underset{\alpha \in [0,1]}{\sup}\underset{P_{pq}:Eh\left(p\right)>0}{\inf}\frac{g(P_{pq},\alpha )}{Eh(p)},$$
where the infimum is taken over distributions $P_{pq}$ of the form $\left(1-\beta \right)Q_{a_{1},a_{2}}+\beta Q_{b_{1},b_{2}}$ with 0≤a≤t<b≤1 and β=0 or $\beta =\frac{t-a}{b-a}>0$ such that Eh(p)>0. (Recall the definition of a,b in (Equation 6)). □

## 5. Discussion

The breakthrough made by Gilmer [7] shows the power of information-theoretic techniques in tackling problems in related fields. In fact, the union-closed sets conjecture has a natural interpretation in the information-theoretic (or coding-theoretic) sense. Consider the memoryless OR multi-access channel $\left(x^{n},y^{n}\right)\in \Omega^{2n}\mapsto x^{n}\vee y^{n}\in \Omega^{n}$. We would like to find a nonempty code $A\subseteq \Omega^{n}$ to generate two independent inputs $X^{n},Y^{n}$ with each following Unif(A) such that the input constraint $E\left[X_{i}\right]\leq t,\forall i\in \left[n\right]$ is satisfied and the output $X^{n}\vee Y^{n}$ is still in *A* a.s. The union-closed sets conjecture states that such a code exists if and only if t≥1/2. Based on this information-theoretic interpretation, it is reasonable to see that the information-theoretic techniques work for this conjecture. It is well-known that information-theoretic techniques usually work very well for problems with “approximate” constraints, e.g., the channel coding problem with the asymptotically vanishing error probability constraint (or the approximate version of the union-closed sets problem introduced in [9]). It is still unclear whether information-theoretic techniques are sufficient to prove sharp bounds for problems with “exact” constraints, e.g., the zero-error coding problem (or the original version of the union-closed sets conjecture).

Furthermore, as an intermediate result, it has been shown that $\Gamma_{n}\left(t\right)>1$ implies $p_{A}>t$ for any OR-closed $A\subseteq \Omega^{n}$. Here $\Gamma_{n}\left(t\right)$ is given in (Equation 8), expressed in the multi-letter form (i.e., the dimension-dependent form). By the super-block coding argument, it is verified that, given t>0, $\lim_{n\to \infty }\Gamma_{n}\left(t\right)$ exists. It is interesting to investigate this limit and prove a single-letter (dimension-independent) expression for it.

For simplicity, in this paper, we only consider the maximal correlation coefficient as the constraint function. In fact, the maximal correlation coefficient used here can be replaced by other functionals. The key property of the maximal correlation coefficient we used in this paper is the “tensorization” property, i.e., (Equation 10) (in fact, only “≤” part of (Equation 10) was used in our proof). In the literature, there is a class of measures of correlation satisfying this property, e.g., the hypercontractivity constant, strong data processing inequality constant, or, more generally, Φ-ribbons, see [20,21,22]. (Although the tensorization property in the literature is only defined and proven for independent random variables, this property can be extended to the coupling constructed in (Equation 9)). Following the same proof steps given in this paper, one can obtain various variants of Theorem 1 with the maximal correlation coefficient replaced by other quantities, as long as these quantities satisfy the tensorization property. Another potential direction is to replace the Shannon entropy with a class of more general quantities, Rényi entropies. However, unfortunately Rényi entropies do not satisfy the chain rule (unlike the Shannon entropy), which leads to a serious difficulty in single-letterizing the corresponding multi-letter bound such as $\Gamma_{n}\left(t\right)$ in (Equation 8) (i.e., in making the multi-letter bound dimension-independent).

## Data Availability

Not applicable.

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
