# Peer review of "Dimension-Free Bounds for the Union-Closed Sets Conjecture"

_entropy, 2023, doi:10.3390/e25050767_

Round 1

Reviewer 1 Report

See attached pdf

Reviewer 2 Report

This paper breaks past a natural barrier of (3-sqrt(5))/2 for a very natural problem, the union-closed conjecture. Although Sawin in an earlier paper gave a similar idea to the one this paper implements, it is clear from the author's previous works (which were, appropriately, in my opinion, cited in the paper under review) that he himself had used the idea before. 

Despite Sawin's idea being optimally analyzed in a more recent paper of Cambie, with a better constant than the one obtained in the paper under review, we recommend the paper be accepted. The reason is that the union-closed conjecture is a very basic problem, with this paper making a nontrivial contribution, written with a greater generality in mind than the other recently published papers. Such generality and other comments written throughout the paper might help further progress on the conjecture.

Frankly, the notation made this paper very difficult (and unappetizing) to read. However, once absorbing the notation, it was not difficult to read. The author uses the same notation in his papers he cited, which is probably the motivation for that notation here, but it is a bit of an unnecessary barrier to breach for this short paper on a pretty accessible conjecture.  

Some more minor comments:

Line 64, I think you want "\rho,t>0" instead of "s,t > 0"

Line 77, Gamma(t) > 1 instead of > 0

Equation (3) - maybe explain this

Line 102 - you use the Shannon entropy symbol H here and then again on Line 104 before defining it on Line 105. 

Line 184 - "natrual" --> "natural"

Round 2

Reviewer 1 Report

The paper looks great now!

Author Response

Thank you for reviewing this manuscript!

Reviewer 3 Report

Congratulations on a nice paper.

Author Response

(The authors gave the same response as above.)
